# Comparison of In-Situ Chlorophyll-*a* Time Series and Sentinel-3 Ocean and Land Color Instrument Data in Slovenian National Waters (Gulf of Trieste, Adriatic Sea)

El Khalil Cherif [1,2,3,*], Patricija Mozetič [2], Janja Francé [2], Vesna Flander-Putrle [2], Jana Faganeli-Pucer [4] and Martin Vodopivec [2,*]

1   Institute for Systems and Robotics, Instituto Superior Técnico, University of Lisbon, 1649-004 Lisbon, Portugal
2   National Institute of Biology, Marine Biology Station Piran, Fornače 41, 6330 Piran, Slovenia; patricija.mozetic@nib.si (P.M.); janja.france@nib.si (J.F.); vesna.FlanderPutrle@nib.si (V.F.-P.)
3   Laboratory of Chemistry Research Unit (CIQUP), Faculty of Sciences, University of Porto, 4099-002 Porto, Portugal
4   Faculty of Computer and Information Science, University of Ljubljana, Večna pot 113, 1000 Ljubljana, Slovenia; jana.faganeli@fri.uni-lj.si
*   Correspondence: elkhalil.cherif@fc.up.pt (E.K.C.); Martin.Vodopivec@nib.si (M.V.); Tel.: +21-266-639-0481 (E.K.C.); +386-59-232-905 (M.V.)

**Abstract:** While satellite remote sensing of ocean color is a viable tool for estimating large-scale patterns of chlorophyll-a (Chl-*a*) and global ocean primary production, its application in coastal waters is limited by the complex optical properties. An exploratory study was conducted in the Gulf of Trieste (Adriatic Sea) to assess the usefulness of Sentinel-3 satellite data in the Slovenian national waters. OLCI (Ocean and Land Colour Instrument) Chl-*a* level 2 products (OC4Me and NN) were compared to monthly Chl-*a* in-situ measurements at fixed sites from 2017 to 2019. In addition, eight other methods for estimating Chl-*a* concentration based on reflectance in different spectral bands were tested (OC3M, OC4E, MedOC4, ADOC4, AD4, 3B-OLCI, 2B-OLCI and G2B). For some of these methods, calibration was performed on in-situ data to achieve a better agreement. Finally, L1-regularized regression and random forest were trained on the available dataset to test the capabilities of the machine learning approach. The results show rather poor performance of the two originally available products. The same is true for the other eight methods and the fits to the measured values also show only marginal improvement. The best results are obtained with the blue-green methods (OC3, OC4 and AD4), especially the AD4SI (a designated fit of AD4) with R = 0.56 and RMSE = 0.4 mg/m$^3$, while the near infrared (NIR) methods show underwhelming performance. The machine learning approach can only explain 30% of the variability and the RMSE is of the same order as for the blue-green methods. We conclude that due to the low Chl-*a* concentration and the moderate turbidity of the seawater, the reflectance provided by the Sentinel-3 OLCI spectrometer carries little information about Chl-*a* in the Slovenian national waters within the Gulf of Trieste and is therefore of limited use for our purposes. This requires that we continue to improve satellite products for use in those marine waters that have not yet proven suitable. In this way, satellite data could be effectively integrated into a comprehensive network that would allow a reliable assessment of ecological status, taking into account environmental regulations.

**Keywords:** chlorophyll-*a*; Sentinel-3; OLCI; machine learning; coastal waters; Gulf of Trieste





## 1. Introduction

Chlorophyll-*a* (Chl-*a*) is the major pigment responsible for photosynthesis and is therefore found in every photosynthesizing organism, including phytoplankton. Despite the bias associated with converting Chl-*a* concentration to organic carbon and the variability of its cell content, Chl-*a* remains the most commonly used proxy for phytoplankton biomass in aquatic systems [1]. Chl-*a* concentration is one of the key indices in the study of

the trophic status of any natural marine ecosystem [2] and for estimating the rate of primary production [3]. Increased concentrations of Chl-*a* in seawater, as a consequence of the elevated nutrient supply, can have detrimental effects on the marine environment. Therefore, Chl-*a* has been recognized as a reliable and easily attainable parameter for the assessment of the ecological status of marine waters in relation to eutrophication pressure [4]. In European countries, its use as an indicator is mandated by European directives (WFD, 2000/60/EU and MSFD, 2008/56/EC) and Regional Sea Conventions, with well-defined classification criteria [5].

In addition to nutrient availability, either from autochthonous or allochthonous sources (e.g., river discharges), phytoplankton biomass is influenced by other environmental factors, such as light, water column temperature and stratification, and wind stress [6–9]; as well as top-down control of grazers [10,11]. In shallow coastal waters, the presence of suspended matter reduces light penetration through the water and affects light availability for photosynthesis and therefore growth of phytoplankton [12].

Conventional monitoring of Chl-*a* concentration, which relies on analytical methods to determine Chl-*a* (spectrophotometry, fluorimetry), provides in-situ data at a limited number of water column depths, locations and with relatively low frequency [6,8,12–15]. Therefore, the use of remote sensing technology has the potential to contribute to this topic significantly. Satellites provide frequent monitoring of large surfaces when they are not obscured by clouds [6,8,16]. On the other hand, satellite-based Chl-*a* estimation is subject to considerable uncertainties in some areas [17].

In recent years, various satellite sensors and models have emerged as useful tools for Chl-*a* estimations in the sea waters [18]. The latter are classified as Case 1 and Case 2. Case 1 waters are those waters whose inherent optical properties (IOPs) are dominated by phytoplankton (e.g., most open ocean waters), whereas Case 2 waters are all other waters (e.g., some coastal and inland waters containing colored dissolved organic matter (CDOM) and inorganic mineral particles in addition to phytoplankton) [19–23]. The blue-green methods have proved to be suitable in Case 1 waters [24], while Chl-*a* estimation is challenging in Case 2 areas. Several different algorithms and methods have been developed for the Chl-*a* concentration estimation in Case 1 and Case 2 waters [24–32].

O'Reilly et al. (1998) developed an ocean color algorithm (OC4) using the blue-green band ratio method [28]. There are many variations of OC4 adapted for different sensors and locations. OC3 is a simplification of OC4 and uses reflectance at three different wavelengths instead of four [33]. These algorithms have been shown to yield accurate results in Case 1 waters [31,34]. On the other hand, the near infra-red (NIR) bands are less affected by bottom reflectance, CDOM and TSM [32,35]. In fact, several studies have used NIR bands for the development of Chl-*a* estimation algorithms in Case 2 waters [31,35–37], such as the two bands (2B-OLCI) and the three bands (3B-OLCI) models [38], as well as the empirical algorithm based on a two-band red-NIR algorithm (G2B) [32].

The Sentinel-3 algal pigment concentration is derived through two different models: OC4Me and Neural Network (NN). These products are free of charge—downloaded from https://eoportal.eumetsat.int/, (accessed on 15 July 2020)—and updated every 24 h. OC4Me is the pigment concentration based on the ocean color for the medium resolution imaging spectrometer (MERIS) algorithm developed by [39]. The NN product is derived through the alternative atmospheric correction processor known as C2RCC. This is a combined atmospheric and ocean neural network approach based on original work by [40].

There are several studies that focus on the performance and sometimes the development of different algorithms for specific geographical areas. These range from Chesapeake Bay in North America [25,37,41,42], South America [35], the Baltic Sea [17], the Mediterranean Sea [43–45], South Africa [46], to Chinese coastal waters [47]. There are also several studies using the remote sensing data for Chl-*a* determination in the Adriatic Sea. In particular, the site of the Acqua Alta Oceanographic Tower (AAO), which is located in front of Venice about 80 km from Slovenian national waters, has had an extensive optical characterization [48–51]. On the other hand, the ecosystem in the western part of the northern

Adriatic differs considerably from the southeastern part of the Gulf of Trieste and to our knowledge, no similar study is available for the Slovenian national waters. Mauri et al., 2007 [52], used moderate resolution imaging spectroradiometer (MODIS) data to relate Chl-*a* concentration with forcing parameters in northern Adriatic. The study included the Gulf of Trieste, but in-situ data were not included. Mozetič et al., 2010 [53], used monthly averaged sea-viewing wide field-of-view sensor (SeaWiFS) data, area-averaged over several larger zones, and also made a comparison with in-situ data.

The high frequency of satellite measurements can provide information about a sudden increase in phytoplankton biomass or the development of a bloom that would go unnoticed with the existing frequency of in-situ sampling, which is usually conducted fortnightly to monthly in monitoring programs. These data may also be useful in validating numerical models and data assimilation. New tools based on remote sensing and modeling can provide new data to complement existing in-situ methods for Chl-*a* determination. In addition, remote sensing methods provide information on a larger, i.e., regional scale with a reduction in time and economic costs.

The aim of this study is to evaluate the possibilities of using Sentinel-3 data for Chl-*a* monitoring in the Slovenian national waters within Gulf of Trieste, Adriatic Sea. The Sentinel-3 OLCI data are available daily and in high spatial resolution (300 m), which should prove very useful in this small area where many monitoring locations are close to the shore.

## 2. Materials and Methods

### 2.1. Study Area and In-Situ Sampling

Slovenian national waters are located in the southeastern part of the Gulf of Trieste (GoT), which forms the northernmost end of the Adriatic Sea (Figure 1) and thus of the Mediterranean Sea. Due to its location, occasional strong winds [54,55] and shallow depth (mostly less than 30 m), it is subject to high temperature variations [54]. It is also a location of considerable freshwater inflow, mainly from the Soča (Isonzo) River and several smaller rivers. In summer, the water is stratified, while in winter the column is well mixed [56].

A 30-year record of in-situ observations of physical, chemical and biological parameters, including phytoplankton biomass and community structure, has shown that not only GoT but the entire northern Adriatic has undergone significant changes [57]. These changes resulted mainly in the meteorologically induced intensification of nutrient inputs from rivers combined with a decrease in the use of phosphates in households and more intensive use of nitrates in agriculture in the catchments of tributaries. The area, previously prone to eutrophication, turned to primarily oligotrophic at the turn of the century [10]. At the same time, a regime shift in phytoplankton community structure was observed [10,58,59].

In-situ sampling was conducted at seven sampling stations in the southeastern part of the GoT during 2017–2019 with a fortnightly (long-term ecological research site, LTER) to monthly frequency (the remaining six). Stations 0DB2, 000K and 00MA are located close to the shore with depths ranging from 15 to 17 m (Figure 1, magenta markers), while 00CZ, 00F2 and 00ZM are positioned further offshore towards the middle of the GoT (depth 21 to 24 m) (Figure 1, blue markers). The LTER site (22 m depth) is the location of the longest ecological time series in the Slovenian part of the GoT. (Figure 1, marked in blue as offshore).

Seawater samples for Chl-*a* analyses were collected using Niskin bottles (5 L) at 0.5 to 1 m depth. Seawater was filtered through GF/F filters and frozen until analysis.

### 2.2. Methodology

In this study, in-situ Chl-*a* concentrations were compared with Chl-*a* concentration estimates based on data from the Sentinel-3 OLCI spectrometer, as shown in flowchart (Figure 2).

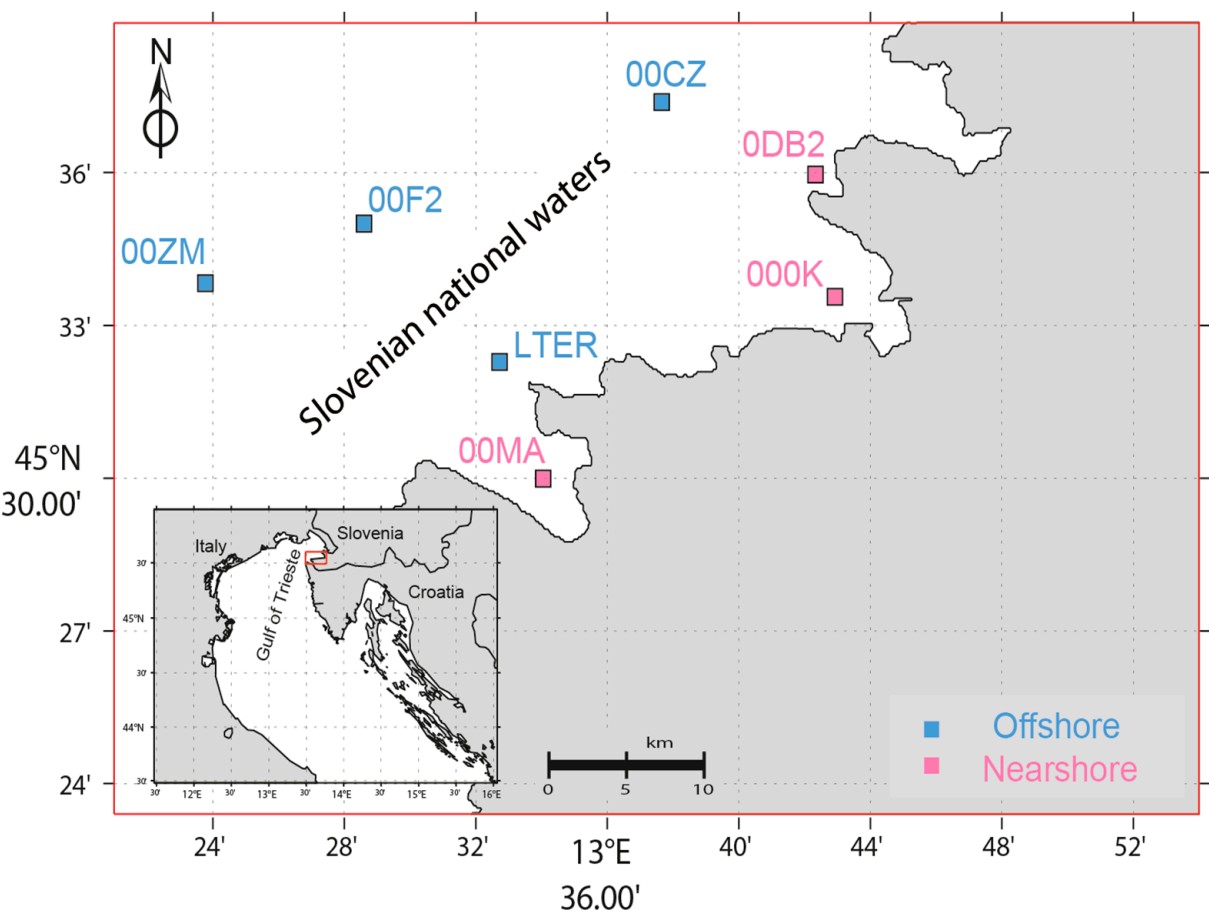

**Figure 1.** Study area and sampling stations in the Slovenian national waters within the Gulf of Trieste, northern Adriatic Sea (inset), which are part of the national monitoring program of the ecological status of the sea.

First, Chl-*a* OLCI level 2 products (OC4Me and NN) were interpolated to sampling stations (using nearest neighbor value) and compared to in-situ measurements (Figure 2). Second, eight known methods for estimating Chl-*a* concentration based on reflectance values were tested (OC3M, OC4E, MedOC4 ADOC4, AD4, 3B-OLCI, 2B-OLCI and G2B). Third, localized calibration was performed for seven known methods (OC3, OC4, AD4, G2B, 2B-OLCI, 3B-OLCI). These were marked with a SI suffix in the acronym (e.g., OC4SI). Fourth, a random forest machine [60,61] was trained on in-situ data.

### 2.2.1. In-Situ Chl-*a* Data: Analytical Methods

Chl-*a* concentrations at sampling stations 0DB2, 000K, 00MA, 00CZ, 00F2 and 00ZM were analyzed fluorometrically in 90% acetone extracts using a laboratory fluorimeter Turner Designs Trilogy [62]. Chl-*a* concentration at the LTER site was determined using a reversed phase high performance liquid chromatography (HPLC) method [63,64]. The HPLC system was equipped with a reversed phase 3 μm C18 column (Pecosphere, 35 × 4.5 mm, Perkin Elmer, Waltham, MA, USA). Chl-*a* was detected by absorbance at 440 nm using a DAD (Agilent Technologies, model 1290 Infinity, Santa Clara, CA, USA). Data acquisition and integration were performed using Agilent ChemStation software.

Comparison of Chl-*a* concentrations determined by both methods on the same samples (data not shown) revealed that the HPLC-derived concentrations were only slightly higher than the fluorometrically determined concentrations (average difference 0.02 mg m$^{-3}$, $R^2 = 0.92$). The good agreement between the two methods justified the use of all data for subsequent analyzes.

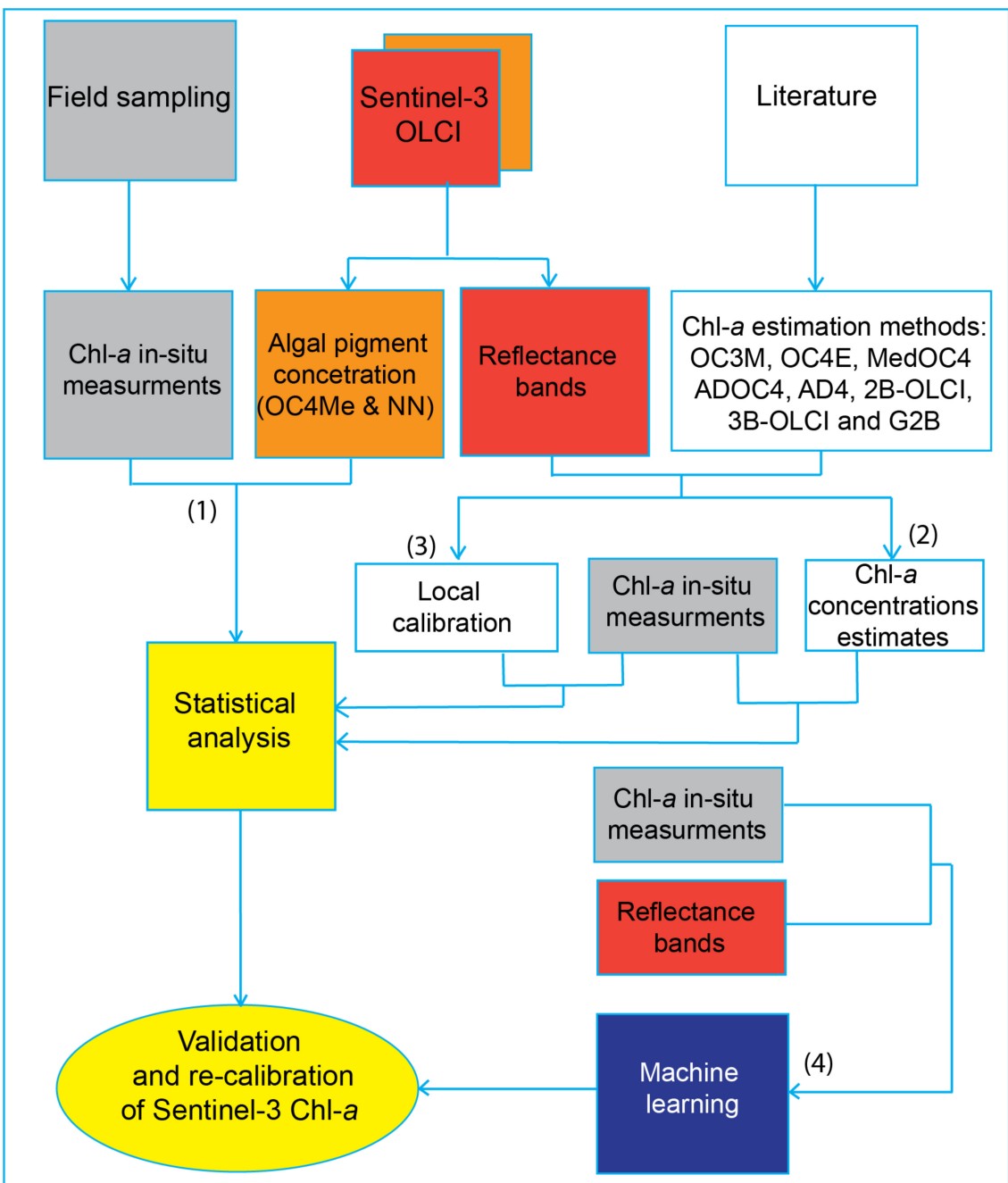

**Figure 2.** Flowchart of the evaluation of Sentinel-3 OLCI data using several different methods. The four important steps of this study are presented: (**1**) Chl-*a* OLCI level 2 products (OC4Me and NN) were evaluated at sampling station locations and compared to in-situ measurements, (**2**) eight known methods for estimating of Chl-*a* concentration based on reflectance values were tested (OC3M, OC4E, MedOC4, ADOC4, AD4, 3B-OLCI, 2B-OLCI and G2B), (**3**) localized calibration was performed for seven known methods (OC3, OC4, AD4, G2B, 2B-OLCI, 3B-OLCI) and (**4**) a L1-regularized regression and a random forests machine were trained on in-situ data.

### 2.2.2. Satellite Data: Chl-*a* Concentration Estimates

A total of 117 time-records (snapshots) of Sentinel-3 OLCI L2 product were used in this study. The images were taken one day before, the same day or one day after the in-situ Chl-*a* measurements. Some of the images were omitted due to cloud obstruction.

The Chl-*a* is usually estimated from satellite multispectral images using semi-empirical algorithms that use different frequency bands (i.e., green and blue part of the visible spectrum) [65]. The normalized spectra [66] used in our study are shown in Figure 3. The analysis was performed on level 2 products obtained from the Sentinel-3 OLCI spectrometer (with a nominal spatial resolution of up to 300 m for OLCI full resolution products), downloaded from https://eoportal.eumetsat.int/ (accessed on 15 July 2020). The OLCI collects data at daily temporal resolution and ESA provides two Chl-*a* products: OC4Me and NN. OC4Me is the pigment concentration based on the ocean color for MERIS using four bands (443, 490, 510 and 560 nm) [39] and follows the approach of O'Reilly et al. (1998) [67] using the blue-green band ratio. The NN product is derived through the alternative atmospheric correction processor known as C2RCC. This is a combined atmospheric and ocean neural network approach [40]. The NN transforms the directional water leaving reflectance measured in eight spectral bands (the results of the MERIS ground segment atmospheric correction procedure), and angles pixel by pixel with high efficiency into three inherent optical properties, i.e., the scattering of all particles b_tsm and the absorption of phytoplankton pigments a_pig. The IOPs (inherent optical properties) b_tsm and a_pig are then converted into concentrations of the water constituents suspended matter and Chl-*a* [40]. The NN is particularly targeted at more complex Case 2 waters where additional constituents must be considered (scattering by total suspended matter, absorption by detrital and Gelbstoff material).

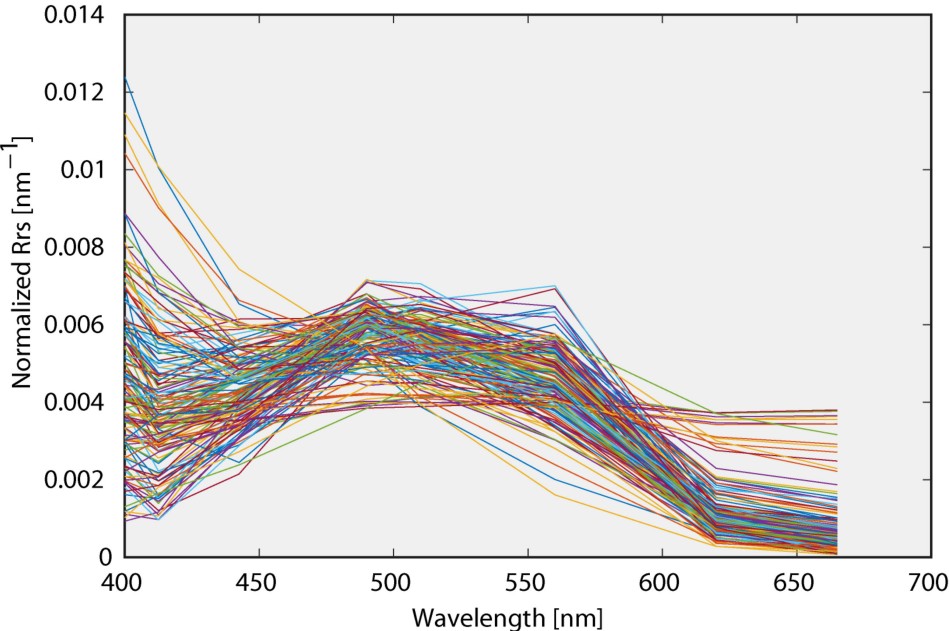

**Figure 3.** The normalized reflectance spectra ($R_{rs}$) used in this study.

Since its launch, the OLCI's performance and associated uncertainties have been under continuous evaluation, especially for nearshore (coastal) waters [25,68–73]. These studies conclude that Chl-*a* remote sensing estimation strongly depends on the chemical composition and hydrodynamic conditions in different areas and demonstrate the importance of in-situ measurements to improve and develop new algorithms.

Several other Chl-*a* methods have been tested as well. Some of these were developed for use in turbid waters and many of them are specialized calibrations. We tested the equations in their original form and in many cases performed a fit to our in-situ data as well. The latter was obtained using MATLAB *fittype*, *fit* and *polyfit* functions and these results are marked with SI in the acronym (e.g., OC4SI, OC3SI). The Sentinel-3 L2 reflectances are already corrected for atmospheric effects and sun specular reflection. A comprehensive list of bands for Sentinels-3 and some of their applications may be

found at https://sentinel.esa.int/web/sentinel/user-guides/sentinel-3-olci/resolutions/radiometric, (accessed on 15 July 2020) and https://sentinel.esa.int/web/sentinel/user-guides/sentinel-3-olci/product-types/level-2-water (accessed on 15 July 2020).

Eight well-known methods for Chl-*a* estimation were used in this part of study, namely OC3E, OC4E, MedOC4, ADOC4, AD4, 2B-OLCI, 3B-OLCI and G2B are presented in details below and in summary in the Appendix A (Table 1).

The OC algorithms are a family of the so-called maximum band ratio algorithms [67]. These were developed to avoid switching between different formulae for different chlorophyll concentrations [33]. They use a maximum ratio of selected bands in the blue and green part of the spectra, and employ a single fourth-order polynomial equation (Chl-*a* concentration is obtained in mg/m³):

$$\text{Chl-a [mg/m}^3] = 10a0 + a1 \times X + a2 \times X2 + a3 \times X3 + a4 \times X4 \tag{1}$$

$$X = \log 10 \text{ (MBR)} \tag{2}$$

where MBR is the maximum ratio of reflectances ($R_{rs}$) in the blue and green part of the spectra.

OC3E:

The first algorithm used in our study uses three different bands. The OC3E [46,74] is formulated for use with MERIS sensor and calibrated on the version 2 of the NASA bio-optical marine algorithm dataset (NOMAD) [75].

$$\text{MBR} = \frac{\max[R_{rs}(443), R_{rs}(490)]}{R_{rs}(560)}$$

The coefficients are: a0 = 0.2521, a1 = −2.2146, a2 = 1.5193, a3 = −0.7702, a4 = −0.4291.

OC4E:

In OC4 algorithms, reflectances at four wavelengths are considered. The OC4Me provided by ESA uses one of these widely used formulations, while the OC4E was adapted for MERIS and calibrated on the same NOMAD dataset as OC3E. Again, the Chl-*a* concentration is obtained in mg/m³ using Equations (1) and (2).

$$\text{MBR} = \frac{\max[R_{rs}(443), R_{rs}(490), R_{rs}(560)]}{R_{rs}(560)}$$

The coefficients are: a0 = 0.3255, a1 = −2.7677, a2 = 2.4409, a3 = −1.1288, a4 = −0.4990.

MedOC4:

The Mediterranean ocean Chlorophyll-a 4 Algorithm (MedOC4) is an empirical algorithm for Chl-*a* retrieval in the Mediterranean Sea [43,50]. This algorithm is developed by using a Mediterranean bio-optical dataset to derive a set of coefficients for a new regional algorithm based on the OC4 functional form using Equations (1) and (2).

$$\text{MBR} = \frac{\max[R_{rs}(443), R_{rs}(490), R_{rs}(510)]}{R_{rs}(560)}$$

The coefficients are: a0 = 0.4900, a1 = −4.023, a2 = 1.428, a3 = 2.976, a4 = −2.795. The latest OLCI specific coefficients were used in our calculations (MedOC4.2020.olci) [76].

ADOC4:

The ADOC4 algorithm is an OC4 algorithm calibrated on the CoASTS dataset [49]. The latter consists of Chl-*a* measurements performed at AAO, which is located in the northern Adriatic Sea. The algorithm is based on SeaWiFS spectrometer data, which slightly differs from OLCI. Instead of the band centered at 555 nm, the closest OLCI band, which is centered at 560 nm, was used.

$$\text{MBR} = \frac{\max[R_{rs}(443), R_{rs}(490), R_{rs}(510)]}{R_{rs}(555)}$$

The ADOC4 coefficients are: a0 = 0.236, a1 = −3.331, a2 = 2.386, a3 = −4.283, a4 = −5.816.

AD4:

As ADOC4, the AD4 is based on the CoASTS dataset as well [77]. In this case, only 2 wavelengths are used and a polynomial of the 3-rd degree was fitted to the data:

$$R = \frac{R_{rs}(490)}{R_{rs}(555)} \tag{3}$$

$$\log 10\ [\text{Chl-}a] = 0.091 - 2.620\ \text{Log}10\ [R_{rs35}] - 1.148\ \text{Log}10\ [R_{rs35}]^2 - 4.949\ \text{Log}10\ [R_{rs35}]^3 \tag{4}$$

AD4 uses the SeaWiFS bands at 490 nm and 555 nm. Again, the OLCI band at 560 nm was used instead of the latter.

2B-OLCI and 3B-OLCI:

The 2B-OLCI is a two-band NIR-red model and the 3B-OLCI is a three-band NIR-red model [78], developed for productive coastal waters of the Azov Sea.

2B-OLCI:

The 2B-OLCI is a two-band NIR-red model [78], developed for productive coastal waters of the Azov Sea:

$$\text{Chl} - a\left[\text{mg/m}^3\right] = 45.597 \times \left(\frac{R_{rs}(708.85)}{R_{rs}(665)}\right) - 26.451 \tag{5}$$

where the remote sensing reflectance ($R_{rs}$) are at 708.85 nm and 665 nm [38].

3B-OLCI:

The 3B-OLCI is a three-band NIR-red model [78], developed for productive coastal waters of the Azov Sea:

$$\text{Chl} - a\left[\text{mg/m}^3\right] = 153 \times \left(\left(\frac{R_{rs}(753.75)}{R_{rs}(665)}\right) - \left(\frac{R_{rs}(753.75)}{R_{rs}(708.75)}\right)\right) + 18.728 \tag{6}$$

where the remote sensing $R_{rs}$ are at 753.75 nm, 708.75 nm and 665 nm [38].

G2B:

G2B is an empirical algorithm based on a two-band red-NIR algorithm [32], developed for application over primarily moderate to high biomass waters [32]:

$$\text{Chl} - a\left[\text{mg/m}^3\right] = \left(\left(35.75 \times \left(\frac{R_{rs}(708.75)}{R_{rs}(665)}\right)\right) - 19.3\right)^{1.124} \tag{7}$$

where the remote sensing $R_{rs}$ are at 708.75 nm and 665 nm [46].

### 2.2.3. Machine Learning

To test if it is possible to infer knowledge from OLCI data we decided to test machine learning models. If state-of-the-art machine learning models are not able to learn how to evaluate Chl-*a* concentrations from OLCI data, we hypothesize that there is just not enough information in OLCI data. We used OLCI reflectance as input parameters for machine learning models. Machine learning models trained on OLCI reflectances have shown good performance in the assessment of Chl-*a* concentrations [47,79,80] at some locations.

We created a dataset containing OLCI reflectances for each day the Chl-*a* concentration was available. So, the target value for modeling was the Chl-*a* concentration for a selected date at a selected sampling site. If the OLCI reflectances were not available for the selected date, the previous or following day's reflectances were taken into account. As there were still a lot of reflectances missing we discarded the dates with too many missing values and reflectances that contained more than one third of missing values. The remaining reflectances were those at 443, 490, 510, 555, 560, 565, 708 and 753.75 nm. Since there were still some missing values in the dataset, we replaced them with median imputation. This means that each missing reflectance was replaced by the median of the reflectances at

the investigated sampling site. We also hypothesized that not all reflectances are useful for the detection of Chl-*a*, so we selected models that incorporate parameter selection simultaneously to fitting the models. We compared the performance of a linear (L1-regularized regression) [81] and a nonlinear model (random forest) [60,61,82]. Before fitting the models, a log transform was applied to inputs and the target value. We applied the log transform because all reflectances and the target value (Chl-*a*) exhibit distributions similar to the log-normal distribution. Most models (especially linear) are suitable for normally distributed values, so we apply a log transform to make the distribution of the values more appropriate for modeling (normal). After using the trained models to make predictions, we applied an exponential transform (inverse of the log transform) to transform the outputs of the model to the same units as the original Chl-*a* values.

### 2.2.4. Statistical Analysis

A basic statistical analysis was performed to relate the satellite-derived Chl-*a* concentration to the in-situ data. The correlation coefficient (R), coefficient of determination ($R^2$) and root-mean-square error (RMSE) were calculated for the satellite-derived Chl-*a* concentrations and their logarithmic values.

## 3. Results and Discussion

ESA algal pigment products (NN and OC4Me), as well as other known Chl-*a* algorithms (OC3E, OC4E, MedOC4, ADOC4, AD4, 2B-OLCI, 3B-OLCI and G2B), and finally, two machine learning methods were tested on in-situ Chl-*a* concentrations. In the following sections, we describe the performance of the tested algorithms for all the stations grouped together. Results for separate nearshore and offshore stations show very similar performance and are presented in the Appendix A (Tables 2 and 3).

### 3.1. The In-Situ Time Series

In-situ Chl-*a* concentrations ranged from 0.09 to 2.52 mg/m$^3$ and exhibited high temporal and spatial variability (Figure 4a). In general, Chl-*a* followed the seasonality known for the GoT with two peaks (in late spring and autumn) [57,83], e.g., peaks in September-November 2017 and November 2018 exceeded 2 mg/m$^3$ at some sampling stations. In contrast, the spring peak was observed only in March and June 2019, when peaks ranged from 1.3 to 1.8 mg/m$^3$. The unusual peak in January 2018 was likely the result of a prolonged high river discharge in late fall-early winter 2017 (http://vode.arso.gov.si/hidarhiv/pov_arhiv_tab.php (accessed on 15 July 2020). However, in many cases there were considerable differences in Chl-*a* concentrations at different stations (e.g., September 2017 and January 2018), reflecting the localized influence of freshwater sources with variable discharge on phytoplankton growth. The observed high spatial variability highlights the importance of high spectrometer spatial resolution.

### 3.2. The Performance of the ESA OLCI Algal Pigment Concentration Products

We start by comparing ESA-provided satellite-derived Chl-*a* concentrations (NN and OC4Me) with in-situ Chl-*a* concentrations. In the LTER area, which represents an offshore station with limited influence from the coasts, the NN algorithm reached two very high values (up to 8 and 11 mg/m$^3$) in October 2017 and January 2018 respectively, which were not reflected in the peaks of the in-situ Chl-*a* concentrations (Figure 4b). Moreover, the highest peaks of in-situ Chl-*a* concentration were only occasionally captured by the NN algorithm. A similar situation in the relationship NN vs. in-situ values was observed at the nearshore station (000K) (Figure 4c). The correlation between the NN Chl-*a* product concentrations and the in-situ measurements was negligible (Figure 5). The correlation coefficient obtained was R = 0.11, and the coefficient of determination was $R^2$ = 0.01 with RMSE = 1.4 (Figure 5a), while the correlation obtained with the logarithmic values was R = 0.31, $R^2$ = 0.00 and RMSE = 0.4 (Figure 5b). These results are somewhat surprising since the NN approach was developed for use in coastal waters.

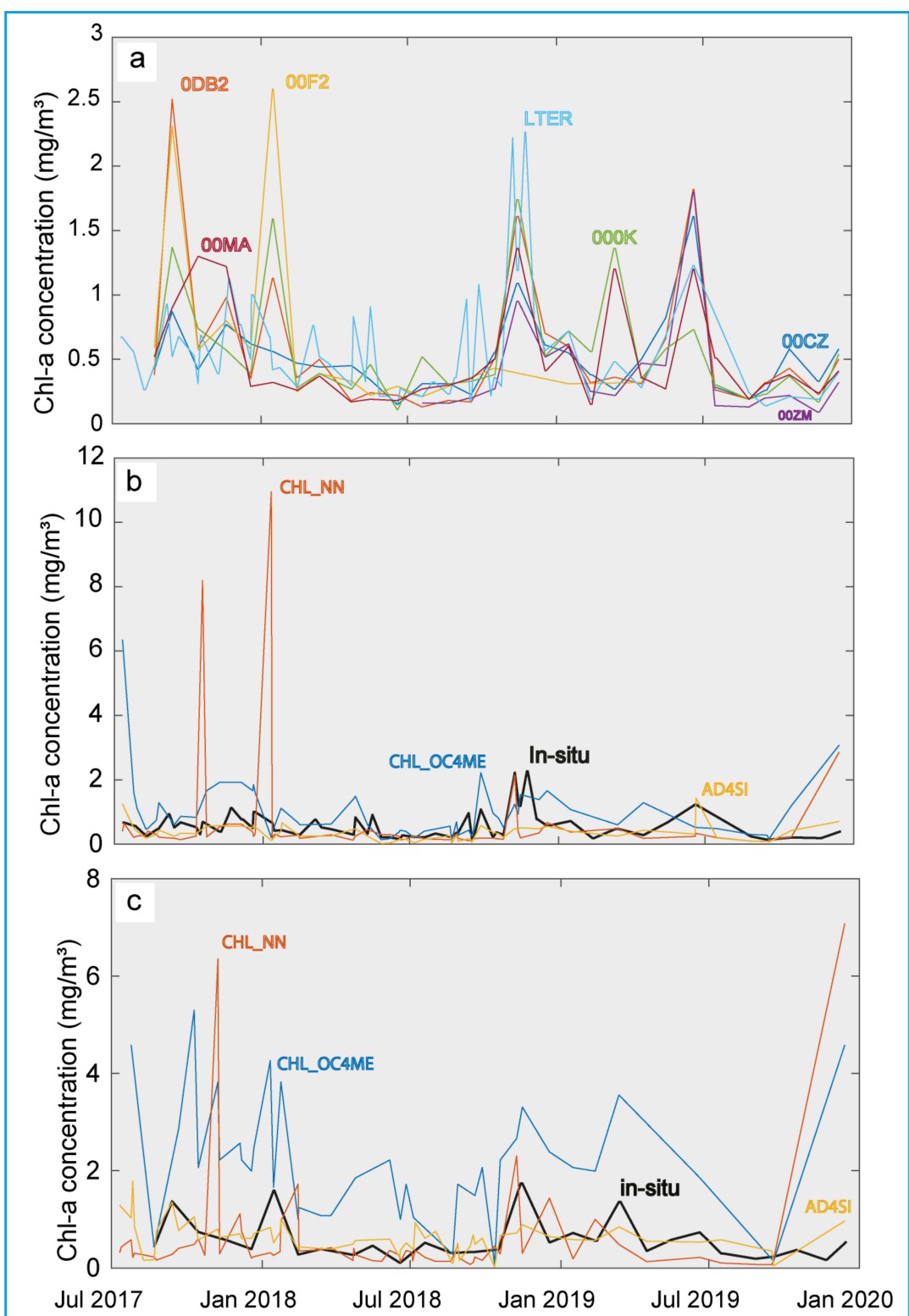

**Figure 4.** (**a**) Time series of in-situ Chl-*a* concentration for all stations considered (see Figure 1). (**b**) Time series for the LTER site. In-situ data are shown in black and selected OLCI-derived concentrations are shown in the respective colors (only values for the dates considered are shown). (**c**) Time series for station 000K. The in-situ data are shown in black and the OLCI products in the respective colors (only values for the dates considered are shown).

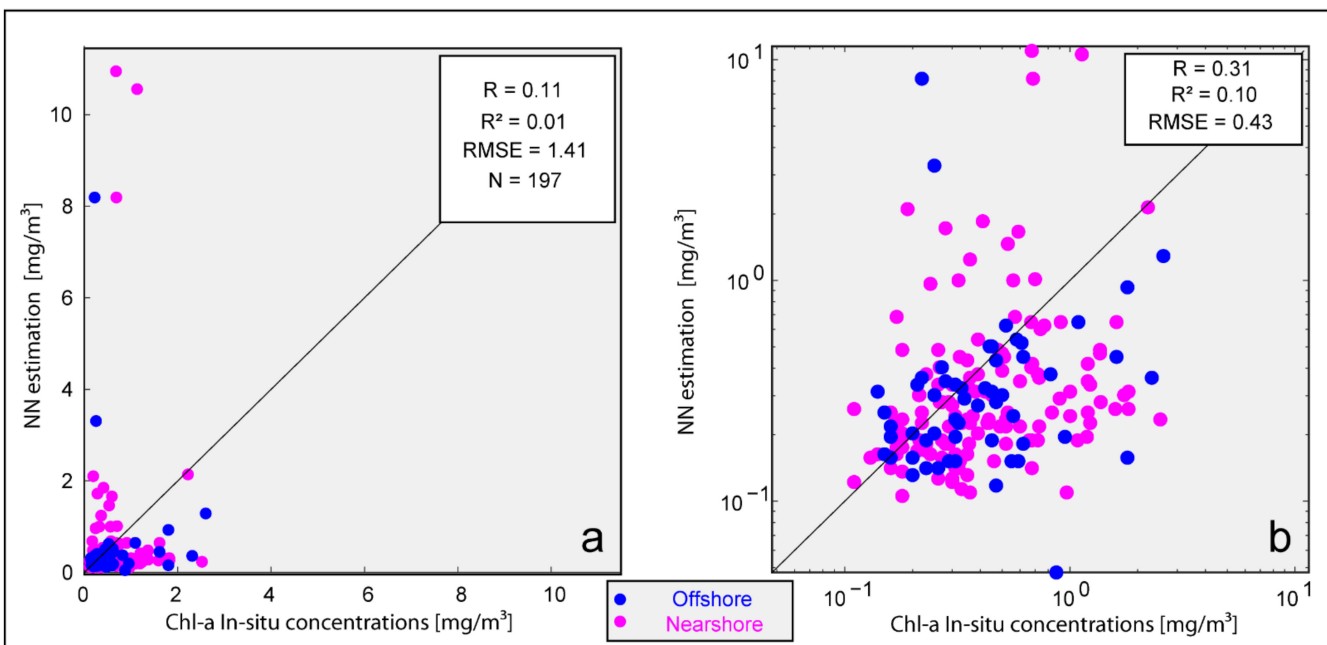

**Figure 5.** (**a**) Scatter plot of in-situ Chl-*a* concentration versus NN Chl-*a* estimates. (**b**) Same in logarithmic scale. The offshore and nearshore stations (Figure 1) are plotted in blue and magenta colors, respectively.

On the other hand, the range of Chl-*a* concentrations measured in-situ and those determined by the OC4Me algorithm was much more similar, but only at the offshore LTER (Figure 4b), while at the nearshore 000K station the OC4Me values were again higher than the measurements (Figure 4c). The differences between these two stations could be explained by the higher turbidity at the 000K station, where the signal detected by the satellite was probably contaminated by suspended particles other than phytoplankton. Indeed, the average Secchi disk depth at the 000K station was $6.0 \pm 2.6$ m during the study period, while it was higher at the LTER station ($9.7 \pm 3.8$ m).

The correlation between the OC4Me and the in-situ Chl-*a* concentration was also better than for NN (Figure 6). The correlation and coefficient of determination for the raw data were R = 0.52 and $R^2$ = 0.27, respectively, with RMSE= 1.1 (Figure 6a), while the correlation obtained with the logarithmic values was R = 0.58, $R^2$ = 0.33 and RMSE = 0.4 (Figure 6b). These results show that there is some information in the blue and green parts of the visible spectrum. The OC4Me values mostly overestimate the measured values (most points are above the black line), which is probably due to reflectance contributed by suspended matter (CDOM) [84].

Even though OC4Me performs better than NN in our case, these results are still rather unsatisfactory. Especially when considering RMSE value of 1.2 mg/m$^3$, which is quite high since the considered measured values are below 3 mg/m$^3$. The availability of many different water-leaving reflectance bands of Sentinel-3 OLCI encourages us towards testing of several other methods in our study area.

### 3.3. The Performance of the Other Available Chl-a Algorithms

A statistical analysis was performed between in-situ Chl-*a* concentrations and several known Chl-*a* algorithms using water leaving reflectance recorded by the Sentinel-3 OLCI spectrometer. We focused on eight known Chl-*a* methods (OC3E, OC4E, MedOC4, ADOC4, AD4, 2B-OLCI, 3B-OLCI and G2B). We note that the same in-situ data were used as in the validation of the OLCI Chl-*a* products (NN and OC4Me).

The best performance was obtained with AD4 [77]. As AD4 was developed in the northern Adriatic, this should be somehow expected. AD4 slightly outperformed the OC4Me with correlation coefficient of 0.53 and root mean square error of 0.7 mg/m$^3$

(Figure 7a). The other algorithms displayed worse performance (Table 1). Those based on green blue bands (OC3M, MedOC4) generally performed better than NIR based algorithms (2B-OLCI, 3B-OLCI, G2B). The Chl-*a* signal in the NIR bands is weaker than in the blue-green part of the spectrum, but less contaminated by suspended matter and bottom reflection. Same results obtained with the same day values only, which are presented in the Appendix A (Table 4). The low Chl-*a* concentrations in Slovenian national waters are probably the cause of the poor performance of these algorithms. AD4 proved to perform best in Slovenian waters, but unfortunately its performance is still unsatisfactory for most applications.

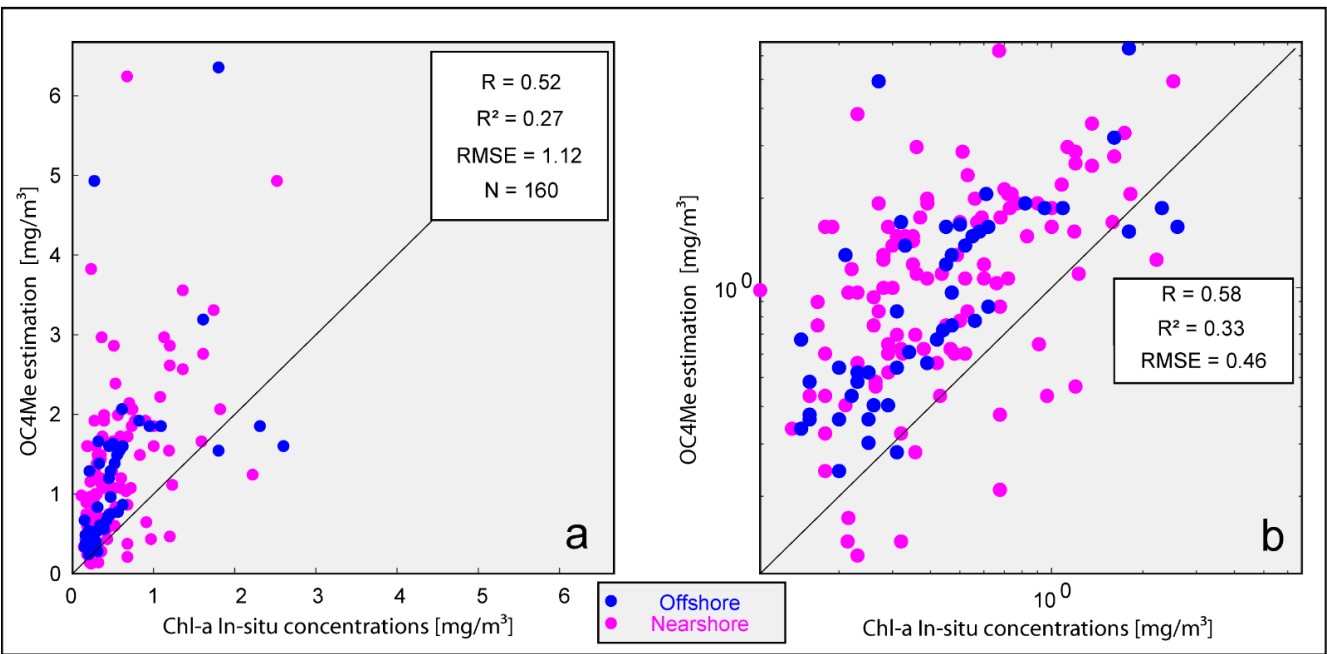

**Figure 6.** (**a**) Scatter plot of in-situ Chl-*a* concentration versus OC4Me Chl-*a* estimates. (**b**) Same in logarithmic scale. The offshore and nearshore stations (Figure 1) are plotted in blue and magenta colors, respectively.

In an effort to get the most out of the reflectance values provided, we performed new calibrations of the existing equations to our in-situ data, which provided new values for the constants involved. These should improve the performance of the Chl-*a* estimation. The fits were performed on the same-day values only. The best result was obtained with the modified AD4 algorithm (AD4SI) recalibrated for the Slovenian national waters, which showed a moderate correlation with R = 0.56 and RMSE = 0.4 mg/m$^3$ (Figure 8). The modified algorithm outperformed the original one, but the correlation is only moderate and RMSE is still quite considerable.

The best fit was obtained with the methods based on the blue and green band ratio. The new coefficients are listed in Table 2.

Returning to the G2B, 3B-OLCI and 2B-OLCI algorithms, which primarily use the NIR bands to determine Chl-*a* concentrations; the poor performance of these methods is likely due to the low concentration of Chl-*a* in the Slovenian national waters. The NIR methods are usually intended for turbid waters and with Chl-*a* concentrations above 10 mg/m$^3$ [32,35]. The Chl-*a* concentrations of our in-situ measurements are below 3 mg/m$^3$ as shown in Figure 9a. Figure 9b shows low to moderate turbidity in the area of interest, with $R_{rs}(709)/R_{rs}(560)$ values mostly below 0.28, which is regarded as the upper limit for clear Case 2 waters [86]. Multi-year observations with SeaWiFS show that turbidity in the Gulf of Trieste alternates between Case 1 and Case 2 depending on environmental conditions [87]. In fact, in-situ turbidity measurements classified the water at LTER as low turbidity Case 2 [88]. For the blue-green algorithms this is still rather turbid as these

are intended for the open ocean (Case 1). Therefore, results may be biased [6] due to the presence of significant sediment concentrations.

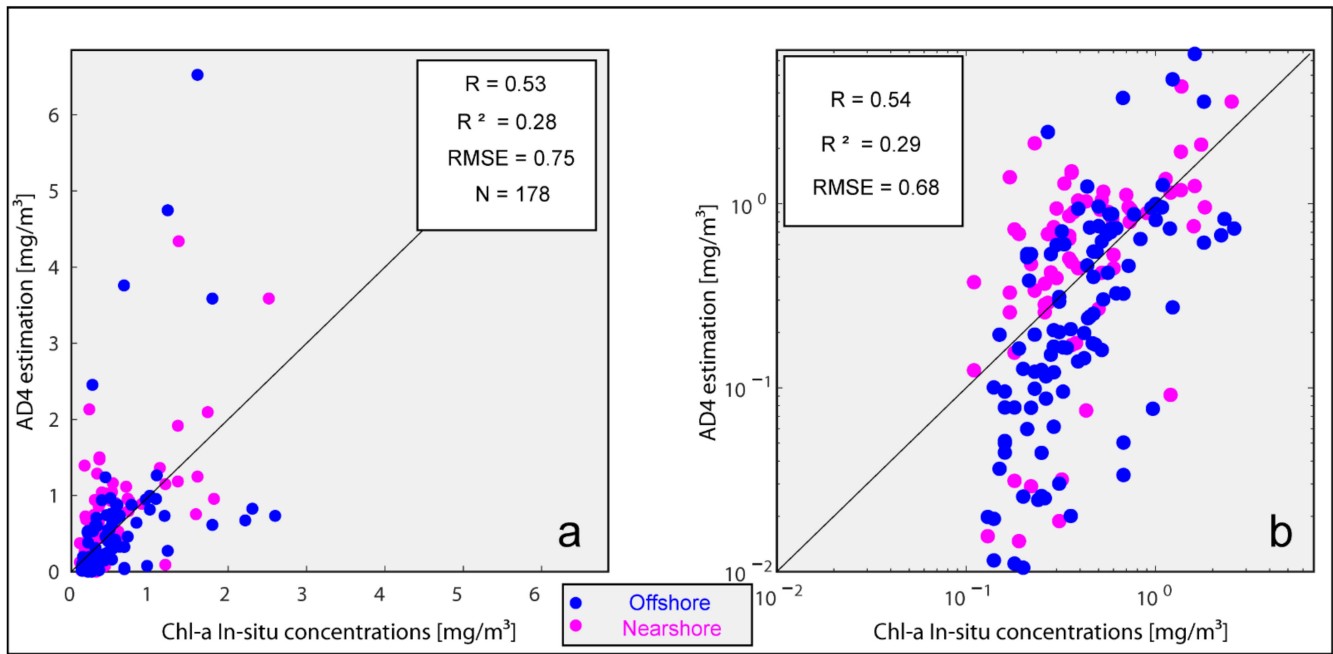

**Figure 7.** (**a**) Scatter plot of in-situ Chl-*a* concentrations versus AD4 Chl-*a* estimates. (**b**) Same in logarithmic scale. The offshore and nearshore stations (Figure 1) are in blue and magenta colors, respectively.

**Table 1.** Statistical analysis (R, R$^2$, RMSE and p-value) for the original (OC4Me, NN, OC3E, OC4E, MedOC4, ADOC4, AD4, 3B-OLCI, 2B-OLCI and G2B) and recalibrated (OC4SI, OC3SI, AD4SI, G2B-SI, 2b-OLCI-SI, 3B-OLCI-S) Chl-*a* algorithms.

| Algorithm Methods | | *n* | R | R$^2$ | RMSE | *p*-Value | R (Log) | R$^2$ (Log) | RMSE (Log) | *p*-Value |
|---|---|---|---|---|---|---|---|---|---|---|
| NN [40] | Original | 197 | 0.11 | 0.01 | 1.4 | 0.1 | 0.31 | 0.10 | 0.43 | $1.10^{-05}$ |
| OC4Me [39,67] | Original | 160 | 0.52 | 0.27 | 1.1 | $3.10^{-12}$ | 0.58 | 0.33 | 0.46 | $2.10^{-15}$ |
| OC4E [75,85] | Original | 179 | 0.52 | 0.27 | 0.9 | $2.10^{-13}$ | 0.59 | 0.35 | 0.42 | $8.10^{-18}$ |
| OC4SI | SI fit | 179 | 0.52 | 0.27 | 0.4 | $1.10^{-13}$ | 0.59 | 0.34 | 0.28 | $1.10^{-17}$ |
| OC3M [67] | Original | 178 | 0.51 | 0.26 | 1.1 | $5.10^{-13}$ | 0.59 | 0.34 | 0.42 | $1.10^{-17}$ |
| OC3SI | SI fit | 178 | 0.52 | 0.27 | 0.4 | $1.10^{-13}$ | 0.58 | 0.34 | 0.27 | $2.10^{-17}$ |
| MedOC4 [43] | Original | 179 | 0.49 | 0.24 | 1.8 | $8.10^{-12}$ | 0.59 | 0.35 | 0.45 | $4.10^{-18}$ |
| G2B [32] | Original | 43 | −0.12 | 0.02 | 32.0 | 0.4 | −0.10 | 0.01 | 1.54 | 0.5 |
| G2B-SI | SI fit | 54 | 0.20 | 0.04 | 0.5 | 0.1 | 0.27 | 0.07 | 0.28 | 0.1 |
| 2B-OLCI [38,78] | Original | 54 | −0.18 | 0.04 | 22.2 | 0.1 | −0.01 | 0.01 | 1.44 | 0.8 |
| 2B-OLCI-SI | SI fit | 54 | −0.10 | - | 0.5 | 0.4 | −0.13 | 0.01 | 0.28 | 0.2 |
| 3B-OLCI [38,78] | Original | 48 | −0.23 | 0.07 | 96.1 | 0.1 | −0.20 | 0.05 | 1.97 | 0.4 |
| 3b-OLCI-SI | SI fit | 48 | 0.25 | 0.07 | 0.5 | 0.1 | 0.27 | 0.08 | 0.28 | 0.1 |
| ADOC4 [49] | Original | 179 | 0.51 | 0.26 | 0.8 | $9.10^{-13}$ | 0.50 | 0.25 | 0.66 | $1.10^{-12}$ |
| AD4 [77] | Original | 178 | 0.53 | 0.28 | 0.7 | $5.10^{-14}$ | 0.54 | 0.29 | 0.68 | $1.10^{-14}$ |
| AD4SI | SI fit | 178 | 0.56 | 0.32 | 0.4 | $6.10^{-16}$ | 0.54 | 0.30 | 0.35 | $7.10^{-15}$ |

The correlation coefficient of AD4 is of the same order of magnitude as the SeaWiFS monthly and area averaged results used in a previous study [53]. In the latter, three stations were used, and the correlation coefficients ranged from R = 0.530 to R = 0.557. The fact that our work, which used higher spatial and temporal resolution and a newer instrument, did not produce better results is somewhat disappointing and further confirms that the performance of remote sensing methods is probably limited by the physical properties of the Slovenian national waters rather than the characteristics of the spectrometer.

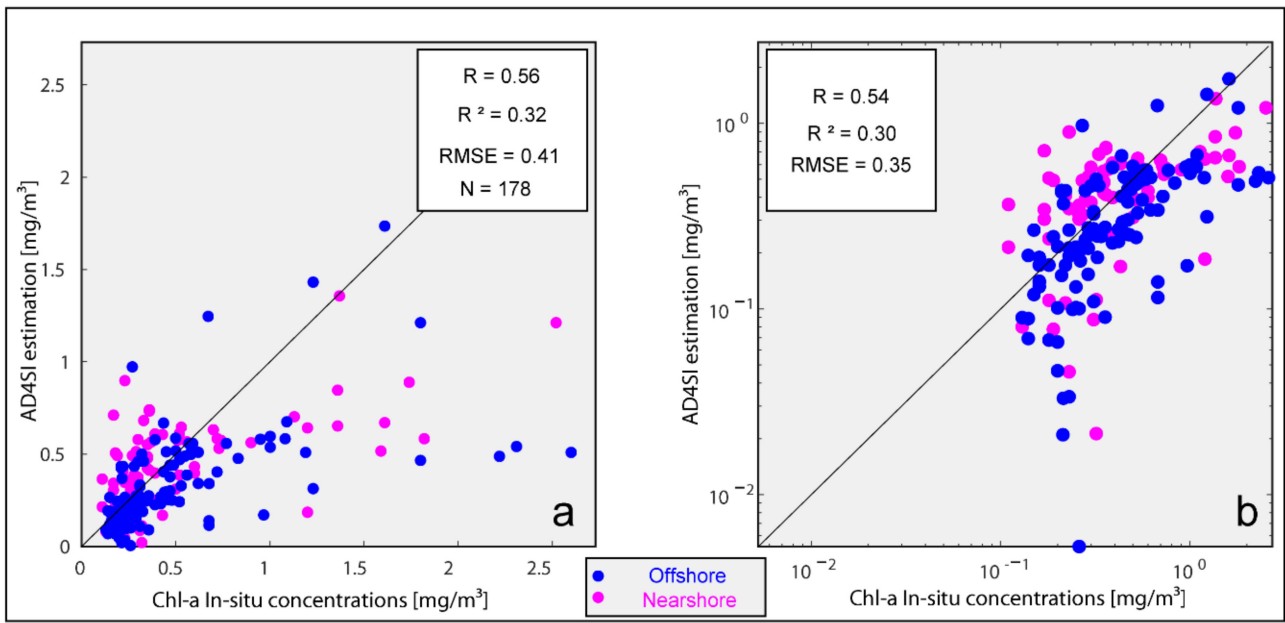

**Figure 8.** (**a**) Scatter plot of in-situ Chl-*a* versus AD4SI. (**b**) Same in logarithmic scale. The offshore and nearshore stations (Figure 1) are in blue and magenta colors, respectively.

**Table 2.** The new coefficients of the OC3SI, OC4SI and AD4SI algorithms.

| Algorithm | a0 | a1 | a2 | a3 | a4 |
|---|---|---|---|---|---|
| OC3SI | −0.10793 | −1.2335 | 1.8051 | −4.5426 | 3.7072 |
| OC4SI | −0.06558 | −1.421 | 2.1237 | −8.1049 | 11.0081 |
| AD4SI | −0.17626 | −1.3869 | −0.17626 | −2.6716 | - |

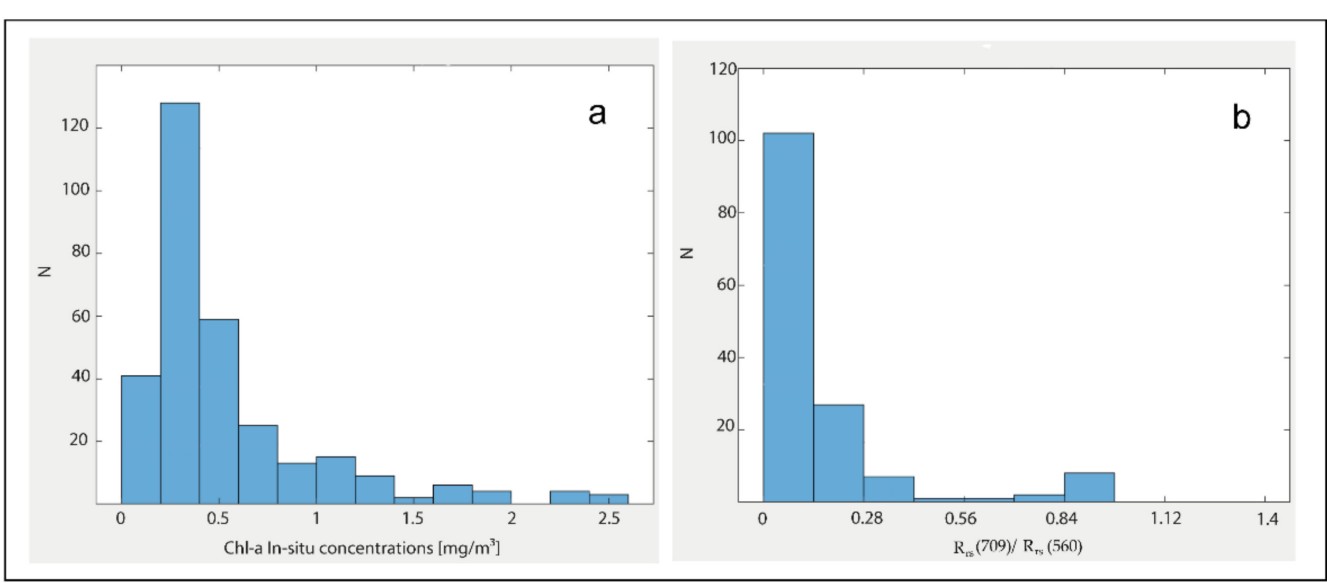

**Figure 9.** Distribution of in-situ (**a**) Chl-*a* concentration and (**b**) turbidity in the southeastern part of the GoT.

### 3.4. Machine Learning

Since the performance of the tested methods was relatively low, we employed two machine learning models to extract the available information from the OLCI data.

We evaluated the performance of the L1-regularized linear regression and random forest models and compared their performance to the baseline model. The baseline model was the model that predicted the median value of the in-situ Chl-*a* from the training set for each day, as is common when evaluating regression models. This helps us to evaluate the inference capability of our models.

The models were only able to explain about 0.24 and 0.22 of variance, respectively (L1-regularized linear regression and random forest model). They were also just barely able to outperform the baseline model (see Figure 10) when evaluated using leave-one-out cross-validation (all but one instances are used to train the model, the remaining one is used for testing and the process is repeated for all instances). To put it in layman's terms, the average in-situ value was a comparably good prediction to the predictions of our models.

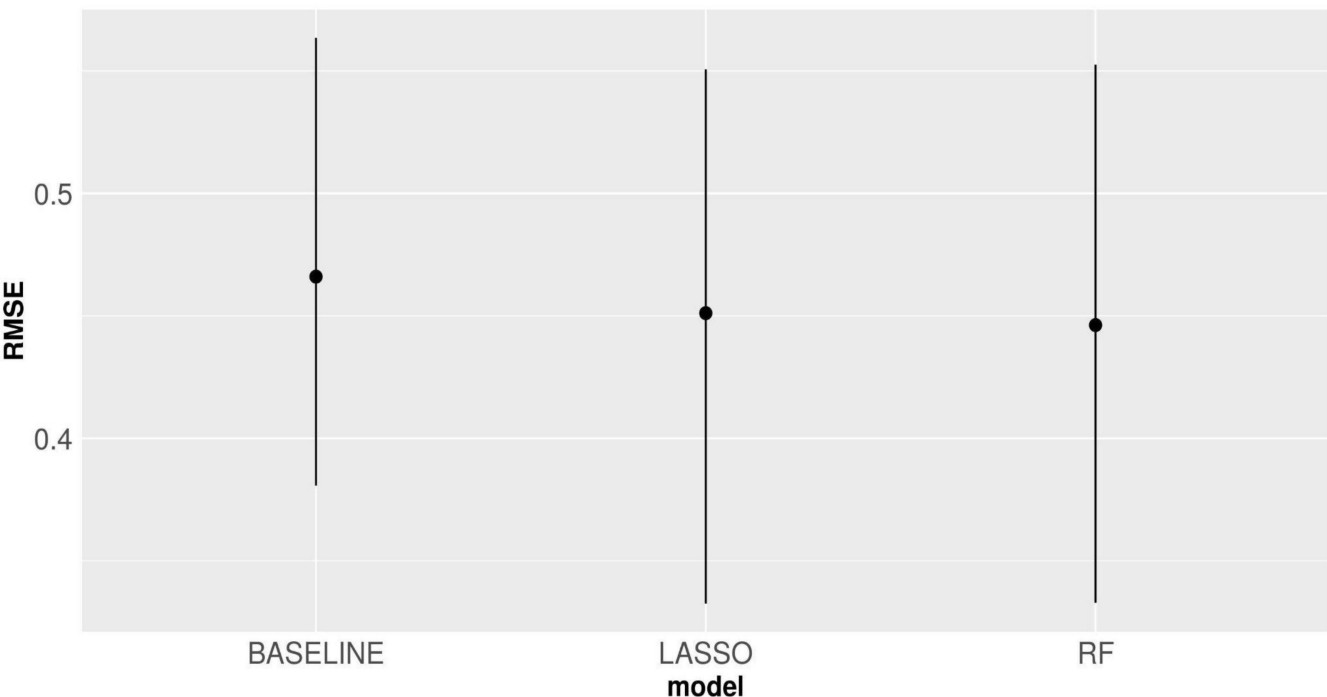

**Figure 10.** RMSE with bootstrap confidence intervals for the baseline, L1-regularized regression (LASSO) and random forest (RF) models.

## 4. Conclusions

In this study, we tested the potential use of Sentinel-3 OLCI imagery for the estimation of Chl-*a* concentrations in the Slovenian national waters (Gulf of Trieste), which could be applied in Slovenian monitoring programs. Two Chl-*a* products provided by ESA (NN, and OC4Me) and concentrations obtained by various other methods (OC3E, OC4E, MedOC4, ADOC4, AD4, 3B-OLCI, 2B-OLCI and G2B) were compared with in-situ measurements.

The methods, which operate in near-infrared (NIR) bands and are originally intended for turbid and productive waters have produced very poor results. As this might be somewhat surprising at first, an overview of the measured Chl-*a* concentration shows that productivity in the area of interest is rather low (0.5 mg/m$^3$ average)—probably too low to leave enough of a footprint in the NIR part of the spectrum.

The blue-green methods (OC3, OC4 and AD4) give better results, despite being developed for the open ocean. Slovenian waters can mostly be characterized somewhere

between the Case 1 and Case 2 waters, which explains why the performance of these methods is still rather poor. The Chl-*a* signal in the blue and green bands is stronger than in the NIR part of the spectrum, but also significantly contaminated by suspended matter and in some cases probably also bottom reflection. Best results were obtained by AD4SI (a designated fit of AD4) with R = 0.56 and RMSE = 0.4 mg/m$^3$.

Finally, in order to extract all available information, machine learning was used with eight OLCI bands as input to an L1-regularized regression and a random forest model, but this also proved unsuccessful. The best of the two methods could only explain 30% of variance and could hardly outperform the baseline model.

Based on the presented results of our study, we can conclude that the Sentinel-3 OLCI data are of very limited use for assessing Chl-*a* concentration in the Slovenian national waters. Productivity is low and turbidity is moderate and therefore the signal does not contain enough information to make reliable Chl-*a* concentration estimates. However, there is always the possibility that the Sentinel-3 OLCI may detect episodes of phytoplankton blooms, which have occurred more frequently in the past but may be triggered suddenly in coastal ecosystems due to increased events of meteorological and hydrological variability in recent years [89].

Satellites are recognized worldwide as important tools for assessing the trophic state of oceans and seas. The use of remote sensing is also envisaged for European seas by Marine Strategy Framework Directive to take into account the large spatial-temporal variability of pelagic habitat when assessing the status of eutrophication and biodiversity. The incorporation of satellite-based Chl-*a* concentrations, as well as multi-sensor products related to phytoplankton functional types [43] into in-situ data using modelling and machine learning techniques was recently proposed by the European Commission [90]. Although our study has shown that satellite instruments currently have limited applicability in coastal seas, it nevertheless contains an important message: sustaining monitoring programs with in-situ measurements of Chl-*a* is of enormous importance. With the improvement of satellite products, we foresee an effective integration of in-situ and satellite data, which will in the future allow a reliable assessment of the ecological status of marine waters, taking into account environmental regulations.

**Author Contributions:** Conceptualization, E.K.C., M.V.; formal analysis, M.V., E.K.C. and J.F.-P.; resources, E.K.C., J.F. and V.F.-P.; writing—original draft preparation, E.K.C.; writing—review and editing, M.V., P.M., J.F., V.F.-P. and J.F.-P.; supervision, M.V. and P.M. All authors have read and agreed to the published version of the manuscript.

**Funding:** This research did not receive any specific grant from funding agencies in the public commercial, or not-for-profit sectors.

**Institutional Review Board Statement:** Not applicable.

**Informed Consent Statement:** Not applicable.

**Data Availability Statement:** Not applicable.

**Acknowledgments:** El Khalil Cherif would like to thank all researchers at National Institute of Biology for their availability and collaboration, especially Patricija Mozetič and Martin Vodopivec. El Khalil Cherif would like to thank Soukaina Kerdad, Hakim Boulaassal and Amina Benoutman for their support and help. Patricija Mozetič and Martin Vodopivec would like to acknowledge the financial support by Slovenian Research Agency (research core funding P1-0237 and postdoctoral project Z7-1884). El Khalil Cherif was supported by Deep Blue Project during his mobility to National Institute of Biology. El Khalil Cherif would like to mention the financial support by FCT with the LARSyS—FCT Project UIDB/50009/2020 and by FCT project VOAMAIS (PTDC/EEI-AUT/31172/2017, 02/SAICT/2017/31172).

**Conflicts of Interest:** The authors declare no conflict of interest.

## Appendix A

**Table 1.** Descriptions of the Chl-*a* algorithms used in this study (OC4Me, NN, OC3E, OC4E, MedOC4, ADOC4, AD4, 3B-OLCI, 2B-OLCI and G2B).

| Algorithm | Description |
|---|---|
| OC3E [46,74] | Algorithm used three-band blue-green reflectance ratio of the following format:<br>$MBR = \frac{Max[R_{rs}(443), R_{rs}(490)]}{R_{rs}(560)}$<br>$X = \log10\ (MBR)$<br>Chl-*a* [mg/m³] $= 10^{a0\ +\ a1\ \times\ X\ +\ a2\ \times\ X\ 2\ +\ a3\ \times\ X\ 3\ +\ a4\ \times\ X\ 4}$, with the coefficients: a0 = 0.2521, a1 = −2.2146, a2 = 1.5193, a3 = −0.7702 and a4 = −0.4291. |
| OC4E | Algorithm used four-band blue-green reflectance ratio of the following format:<br>$MBR = \frac{Max[R_{rs}(443), R_{rs}(490), R_{rs}(560)]}{R_{rs}(560)}$<br>Chl-*a* is determined in the same equation as OC3E using the following coefficients:<br>a0 = 0.3255, a1 = −2.7677, a2 = 2.4409, a3 = −1.1288 and a4 = −0.4990. |
| MedOC4 [43,50] | Algorithm is developed by using a Mediterranean bio-optical dataset to derive a set of coefficients for a new regional algorithm based on the OC4 functional form using four-band blue-green reflectance ratio of the following format:<br>$MBR = \frac{MBR[R_{rs}(443), R_{rs}(490), R_{rs}(510)]}{R_{rs}(560)}$<br>Chl-*a* is determined in the same equation as OC3E using the following coefficients:<br>a0 = 0.4900, a1 = −4.023, a2 = 1.428, a3 = 2.976 and a4 = -2.795. |
| ADOC4 [49] | Algorithm used four-band blue-green reflectance of the following format:<br>$MBR = \frac{Max[R_{rs}(443), R_{rs}(490), R_{rs}(510)]}{R_{rs}(555)}$<br>ADOC4 Chl-*a* is determined in the same equation as OC3E using the following coefficients:<br>a0 = 0.236, a1 = −3.331, a2 = 2.386, a3 = −4.283 and a4 = −5.816. |
| AD4 [77] | Algorithm used 2 wavelengths a polynomial of the 3rd degree was fitted to the data using the following format:<br>$R_{rs35} = \frac{R_{rs}(490)}{R_{rs}(555)}$<br>$\log10\ [Chl\text{-}a] = 0.091 - 2.620\ Log10\ [R_{rs35}] - 1.148\ Log10\ [R_{rs35}]^2 - 4.949\ Log10\ [R_{rs35}]^3$ |
| 3B-OLCI [78] | Algorithm used the following bands and formats:<br>Chl-*a* [mg/m³] $= 153 \times ((\frac{R_{rs}(753.75)}{R_{rs}(665)}) - (\frac{R_{rs}(753.75)}{R_{rs}(708.75)})) + 18.728$ |
| 2B-OLCI [78] | Algorithm used the following bands and formats:<br>Chl-*a* [mg/m³] $= 45.597 \times (\frac{R_{rs}(708.85)}{R_{rs}(665)}) - 26.451$ |
| G2B [32] | Algorithm used two-band red-NIR according to the following format:<br>Chl-*a* [mg/m³] $= ((35.75 \times (\frac{R_{rs}(708.75)}{R_{rs}(665)})) - 19.3)^{1.124}$ |

**Table 2.** Statistical analysis (R, $R^2$, RMSE and p-value) for the original (OC4Me, NN, OC3E, OC4E, MedOC4, ADOC4, AD4, 3B-OLCI, 2B-OLCI and G2B) and recalibrated (OC4SI, OC3SI, AD4SI, G2B-SI, 2b-OLCI-SI, 3B-OLCI-S) Chl-*a* algorithms at nearshore points.

| Algorithm Methods | | *n* | | | | | Nearshore | | | |
|---|---|---|---|---|---|---|---|---|---|---|
| | | | R | $R^2$ | RMSE | *p*-Value | R (Log) | $R^2$ (Log) | RMSE (Log) | *p*-Value (Log) |
| NN | Original | 85 | 0.16 | 0.03 | 1.2 | 0.1 | 0.30 | 0.10 | 0.4 | 0.0 |
| OC4Me | Original | 58 | 0.66 | 0.44 | 1.2 | $1.10^{-8}$ | 0.54 | 0.29 | 0.5 | $1.10^{-5}$ |
| OC4E | Original | 71 | 0.61 | 0.37 | 0.9 | $1.10^{-8}$ | 0.55 | 0.31 | 0.4 | $3.10^{-7}$ |
| OC4SI | SI fit | 71 | 0.62 | 0.38 | 0.4 | $6.10^{-8}$ | 0.58 | 0.33 | 0.2 | $2.10^{-7}$ |
| OC3E | Original | 71 | 0.63 | 0.39 | 1.0 | $7.10^{-9}$ | 0.56 | 0.31 | 0.4 | $5.10^{-7}$ |
| OC3SI | SI fit | 71 | 0.61 | 0.38 | 0.4 | $1.10^{-8}$ | 0.55 | 0.31 | 0.3 | $7.10^{-7}$ |
| MedOC4 | Original | 71 | 0.61 | 0.37 | 1.7 | $1.10^{-8}$ | 0.57 | 0.32 | 0.5 | $3.10^{-7}$ |
| G2B | Original | 27 | −0.24 | 0.07 | 39.3 | 0.2 | −0.23 | 0.06 | 1.6 | 0.2 |
| G2B-SI | SI fit | 29 | 0.34 | 0.12 | 0.4 | 0.0 | 0.37 | 0.14 | 0.2 | 0.0 |
| 2B-OLCI | Original | 29 | −0.32 | 0.12 | 28.7 | 0.0 | −0.20 | 0.05 | 1.60 | 0.2 |
| 2B-OLCI-SI | SI fit | 29 | −0.41 | 0.18 | 0.4 | 0.0 | −0.42 | 0.19 | 0.2 | 0.0 |

**Table 2.** *Cont.*

| Algorithm Methods | | $n$ | Nearshore | | | | | | | |
|---|---|---|---|---|---|---|---|---|---|---|
| | | | R | $R^2$ | RMSE | $p$-Value | R (Log) | $R^2$ (Log) | RMSE (Log) | $p$-Value (Log) |
| 3B-OLCI | Original | 27 | −0.35 | 0.14 | 125.2 | 0.0 | −0.20 | 0.05 | 2.0 | 0.4 |
| 3b-OLCI-SI | SI fit | 27 | 0.37 | 0.14 | 0.4 | 0.1 | 0.34 | 0.12 | 0.2 | 0.1 |
| ADOC4 | Original | 71 | 0.61 | 0.38 | 0.7 | $1.10^{-8}$ | 0.47 | 0.22 | 0.5 | $5.10^{-5}$ |
| AD4 | Original | 71 | 0.63 | 0.40 | 0.6 | $5.10^{-9}$ | 0.46 | 0.22 | 0.5 | $5.10^{-5}$ |
| AD4SI | SI fit | 71 | 0.60 | 0.35 | 0.4 | $5.10^{-8}$ | 0.47 | 0.22 | 0.3 | $3.10^{-5}$ |

**Table 3.** Statistical analysis (R, $R^2$, RMSE and p-value) for the original (OC4Me, NN, OC3E, OC4E, MedOC4, ADOC4, AD4, 3B-OLCI, 2B-OLCI and G2B) and recalibrated (OC4SI, OC3SI, AD4SI, G2B-SI, 2b-OLCI-SI, 3B-OLCI-S) Chl-*a* algorithms at offshore points.

| Algorithm Methods | | $n$ | Offshore | | | | | | | |
|---|---|---|---|---|---|---|---|---|---|---|
| | | | R | $R^2$ | RMSE | $p$-Value | R (Log) | $R^2$ (Log) | RMSE (Log) | $p$-Value (Log) |
| NN | Original | 112 | 0.09 | 0.01 | 1.5 | 0.3 | 0.30 | 0.10 | 0.4 | 0.0 |
| OC4Me | Original | 102 | 0.43 | 0.18 | 1.0 | $8.10^{-6}$ | 0.60 | 0.36 | 0.4 | $4.10^{-11}$ |
| OC4E | Original | 108 | 0.46 | 0.21 | 0.9 | $6.10^{-7}$ | 0.62 | 0.38 | 0.3 | $9.10^{-13}$ |
| OC4SI | SI fit | 108 | 0.49 | 0.24 | 0.4 | $1.10^{-7}$ | 0.61 | 0.37 | 0.2 | $4.10^{-12}$ |
| OC3E | Original | 107 | 0.46 | 0.21 | 1.2 | $8.10^{-7}$ | 0.63 | 0.39 | 0.3 | $8.10^{-13}$ |
| OC3SI | SI fit | 107 | 0.49 | 0.24 | 0.4 | $1.10^{-7}$ | 0.62 | 0.38 | 0.2 | $1.10^{-12}$ |
| MedOC4 | Original | 108 | 0.42 | 0.17 | 1.8 | $9.10^{-6}$ | 0.63 | 0.40 | 0.4 | $3.10^{-13}$ |
| G2B | Original | 26 | 0.48 | 0.23 | 11.8 | 0.1 | 0.31 | 0.10 | 1.2 | 0.2 |
| G2B-SI | SI fit | 25 | −0.03 | 0.01 | 0.5 | 0.8 | 0.01 | $3.10^{-6}$ | 0.2 | 1.00 |
| 2B-OLCI | Original | 25 | 0.05 | 0.01 | 10.3 | 0.8 | 0.30 | 0.09 | 1.1 | 0.2 |
| 2B-OLCI-SI | SI fit | 25 | 0.02 | 0.000 | 0.5 | 0.9 | −0.01 | 0.00 | 0.2 | 0.9 |
| 3B-OLCI | Original | 21 | −0.09 | 0.01 | 30.5 | 0.7 | 1.00 | 1.00 | 0.8 | NAN |
| 3b-OLCI-SI | SI fit | 21 | 0.09 | 0.01 | 0.6 | 0.7 | 0.03 | 0.00 | 0.2 | 0.9 |
| ADOC4 | Original | 108 | 0.45 | 0.20 | 0.8 | $1.10^{-6}$ | 0.53 | 0.28 | 0.7 | $4.10^{-9}$ |
| AD4 | Original | 107 | 0.48 | 0.23 | 0.8 | $2.10^{-7}$ | 0.60 | 0.36 | 0.7 | $1.10^{-11}$ |
| AD4SI | SI fit | 107 | 0.56 | 0.31 | 0.4 | $7.10^{-10}$ | 0.60 | 0.36 | 0.3 | $1.10^{-11}$ |

**Table 4.** Statistical analysis (R, $R^2$, RMSE and p-value) for the original (OC4Me, NN, OC3E, OC4E, MedOC4, ADOC4, AD4, 3B-OLCI, 2B-OLCI and G2B) and recalibrated (OC4SI, OC3SI, AD4SI, G2B-SI, 2b-OLCI-SI, 3B-OLCI-S) Chl-*a* algorithms on same-day values.

| Algorithm Methods | | $n$ | R | $R^2$ | RMSE | $p$-Value | R (Log) | $R^2$ (Log) | RMSE (Log) | $p$-Value |
|---|---|---|---|---|---|---|---|---|---|---|
| NN | Original | 75 | 0.03 | 0.00 | 1.6 | 0.8 | 0.15 | 0.02 | 0.4 | 0.2 |
| OC4Me | Original | 56 | 0.57 | 0.32 | 0.8 | $4.10^{-6}$ | 0.54 | 0.29 | 0.4 | $2.10^{-5}$ |
| OC4E | Original | 65 | 0.45 | 0.21 | 0.8 | 0.0 | 0.54 | 0.29 | 0.3 | $4.10^{-6}$ |
| OC4SI | SI fit | 65 | 0.47 | 0.22 | 0.4 | $9.10^{-5}$ | 0.54 | 0.30 | 0.2 | $2.10^{-6}$ |
| OC3E | Original | 64 | 0.53 | 0.28 | 0.9 | $6.10^{-6}$ | 0.56 | 0.31 | 0.3 | $1.10^{-6}$ |
| OC3SI | SI fit | 64 | 0.54 | 0.29 | 0.4 | $5.10^{-6}$ | 0.56 | 0.32 | 0.2 | $1.10^{-6}$ |
| MedOC4 | Original | 65 | 0.43 | 0.19 | 1.6 | 0.0 | 0.54 | 0.29 | 0.4 | $4.10^{-6}$ |
| G2B | Original | 14 | −0.16 | 0.03 | 45.5 | 0.5 | −0.22 | 0.05 | 1.5 | 0.4 |
| G2B-SI | SI fit | 18 | 0.04 | 0.00 | 0.4 | 0.8 | 0.13 | 0.02 | 0.2 | 0.6 |
| 2B-OLCI | Original | 18 | −0.04 | 0.00 | 30.3 | 0.8 | −0.20 | 0.04 | 1.4 | 0.4 |
| 2B-OLCI-SI | SI fit | 18 | 0.12 | 0.01 | 0.4 | 0.6 | 0.06 | 0.00 | 0.2 | 0.8 |
| 3B-OLCI | Original | 15 | −0.14 | 0.02 | 141.6 | 0.6 | −0.61 | 0.37 | 2.3 | 0.2 |
| 3b-OLCI-SI | SI fit | 15 | 0.14 | 0.02 | 0.4 | 0.6 | 0.21 | 0.04 | 0.2 | 0.4 |
| ADOC4 | Original | 65 | 0.45 | 0.20 | 0.7 | 0.0 | 0.51 | 0.26 | 0.4 | $1.10^{-5}$ |
| AD4 | Original | 64 | 0.55 | 0.31 | 0.6 | $1.10^{-6}$ | 0.58 | 0.33 | 0.5 | $6.10^{-7}$ |
| AD4SI | SI fit | 64 | 0.56 | 0.31 | 0.4 | $1.10^{-6}$ | 0.58 | 0.34 | 0.3 | $5.10^{-7}$ |

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
