# Peer review of "Comparison of In-Situ Chlorophyll-a Time Series and Sentinel-3 Ocean and Land Color Instrument Data in Slovenian National Waters (Gulf of Trieste, Adriatic Sea)"

_water, doi:10.3390/w13141903_

Round 1

Reviewer 1 Report

This study compared several Chl-a estimation models based on OLCI data in the Gulf of Trieste. However, the results did not perform well. The reason of the unsuccessful results should be discussed. I don’t think 10 OLCI bands were used as input to machine learning is a good choice. One main concern is the performance of atmospheric correction. I think this is the key step in retrieving Chl-a using ocean color data. And, the logic of the manuscript should be improved. For example, it looks better if section 2.2.3 (Statistical analysis) follows Section 2.3 (Machine learning).

Specific comments:

  1. Line 345, “…the models. a log transform was…” should be “,”.
  2. Figure 3 looks very complicated and not clear.

Reviewer 2 Report

General comments: The manuscript by El Khalil Cherif et al. is devoted to an important issue - to assess the possibilities of using Sentinel-3 space images for monitoring the concentration of chlorophyll-a in seawater using the example of the northern part of the Adriatic Sea. The relevance and importance of these studies are due to the changes in the ecological state of water bodies under the influence of natural and anthropogenic factors. Remote sensing methods are increasingly being used in these studies. The authors have developed a research methodology. In-situ measurements of chlorophyll-a concentration were carried out at 4 offshore and 3 coastal sites for the period from 2017 to 2019. Comparison of in-situ data with the data obtained from Sentinel-3 images was carried out. The results of the work performed have convincingly shown the inefficiency and inexpediency of using satellite data for the task under consideration, at least for the studied water area. Overall, I believe the manuscript is reasonably well written and, in particular, is very detailed in the context of the literature. Basically, I believe the methodology and presentation of the results are sound. However, the presentation of the results in some cases has shortcomings and remarks, which requires minor revision of the manuscript. Specific details are given below.

Comments:

  1. Lines 86-87: The abbreviations NN and MERIS entered here are not decrypted. It is proposed to decrypt.
  2. Fig. 1. The scale is not indicated on the map-scheme Fig.1.
  3. Fig. 2. Explain in the text how the “Validation and improvement of Sentinel-3 Chl-a” procedure is performed.
  4. Lines 224-252: Equations (1), (3) and (5) are the same. (3) and (5) should be deleted and equation (1) should be referenced in the text. The same should be done with equations (2) and (4).
  5. Line 345: It is not explained for what purpose the log transformation was applied. It should be explained in more detail.

Reviewer 3 Report

Review of water-1243431 “Comparison of in-situ chlorophyll-a time series and Sentinel-3 2 Ocean and Land Colour Instrument data in coastal sea (Gulf of 3 Trieste, Adriatic Sea)”

Summary:

Cherif et al. explored the application of satellite remote sensing of ocean colour in coastal waters, in particular the gulf of Trieste, using two available products (OC4ME and NN) and eight estimation methods. Cherif et al.’s results demonstrated a general poor performance of OC4ME, NN and the eight methods. Cherif et al. concluded that low Chl-a concentration together with the moderate turbidity of the seawater in the Gulf of Trieste, could limit the use of remote sensing in coastal waters, raising authors’ concerns about the suitability of this approach as proposed for monitoring programs in two main European Directives.

General comment:

The hypotheses, methodology, results, discussion and conclusion are generally well presented, except for some sections that need to be improved. The manuscript needs some minor changes before being accepted. Details are listed below:

Abstract

Line 35-38: please consider two aspects: 1) WFD and MSFD include several types of water body where remote sensing can be applicable; 2) WFD and MSFD do not reply only on remote sensing for monitoring programs and finally you study is limited to a single (very small) area Gulf of Trieste.  For these reason I would suggest rephrasing the sentence.

Introduction

Line 58: “as well as top-down control of grazers” has different font, please check and add some references.

Lines 68-84: please introduce what are Case 1 and Case 2, and add clearly authors/study that are referring to them.

Lines 85-86: add full names for abbreviations OC4Me and NN before using them, as well as for MERIS

Line 94: since Adriatic Sea is part of the Mediterranean Sea that you have mentioned in the list of study already performed I would suggest adding more references instead of only one.

Line 95: “very few studies” please add their references and description. Otherwise it is not clear the reason of you interest and study

Line 97: please add full name for acronym MODIS, and please put a in italic in Chl-a as done so far

Line 99: please add full name for acronym SeaWiFS

Lines 99-100: you study appears having similar aim to the published Mozetič et al., 2010, I would expected a discussion and/or a reason why your study has been performed

Line 106: please put a in italic in Chl-a as done so far

Line 110: please substitute Slovenian Sea with Gulf of Trieste, Adriatic Sea, as mentioned in the title as well as in the previous paragraphs. Otherwise the reader might get confused. Another option could be Slovenian national waters within Gulf of Trieste, Adriatic Sea

Results and Discussion

Lines 357-358: please put a in italic in Chl-a as done so far

Lines 379-380: please put a in italic in Chl-a as done so far

Lines 382-387 (+Figure 4): please add the p-value

Lines 397-401 (+Figure 5): please add the p-value

Lines 425: Please replace by “Table 1” since table 1 is just in the appendix and not in the main text as for the other tables

Lines 434-436 (Table 2): please add the p-value. Please replace by “Table 1” since table 1 is just in the appendix and not in the main text as for the other tables. Please add in the table caption the full names for all abbreviation used in the table

Lines 440-441: please substitute Slovenian Sea with Gulf of Trieste, Adriatic Sea, as mentioned in the title as well as in the previous paragraphs. Otherwise the reader might get confused. Another option could be Slovenian national waters within Gulf of Trieste, Adriatic Sea

Lines 440-442 (+Figure 7): please add the p-value

Line 471: please use lower case for in-situ

Lines 448: Please replace by “Table 1” since table 1 is just in the appendix and not in the main text as for the other tables

Lines 455+460: Please replace by “Table 2” since table 1 is just in the appendix and not in the main text as for the other tables

Conclusions

Line 503: please replace by “concentrations in Slovenian national waters in the Gulf of Trieste”

Lines 504-506: please clearly mention the two + eight products has done so far

Lines 538: please add author “by”?

Lines 531-540: here as well in the discussion is not mentioned the authors’ concerns about WFD and MSFD that have been clearly state in the abstract (see lines 35-38). Please add your concerns in the discussion/conclusion and/or rephrase/delete statement in the abstract

Supplementary materials

Lines 558-559 (Table 1): please add the p-value and add in the table caption the full names for all abbreviation used in the table

Reviewer 4 Report

See attached file.

Round 2

Reviewer 4 Report

Review of the article “Comparison of in-situ chlorophyll-a time series and Sentinel-3 2 Ocean and Land Colour Instrument data in Slovenian national 3 waters (Gulf of Trieste, Adriatic Sea)”, by Cherif et al. (Revision 1)

I appreciate that the Authors took my comments into consideration. I believe the manuscript is improved. This time have a small list of comments that need attention before making it public.

Lines 72-76: wrong definition of case 1/case 2. It’s about the covariance of these constituents what defines case 1 and case 2, not their absolute concentrations.

Line 90: free of charge if you download them from which place?

Lines 124-125: you already said that

Line 144: define “LTER” before

Line 165: there is something wrong at the beginning of this list within the parentheses

Line 167: need to explain “random forest”

The variable Rrs needs subscript in “rs”

Vertical axes of Figure 3 need units

Line 205: green band of MERIS is 560 nm, not 555 nm.

Line 251: The formula of the MBR [Rrs(443) > Rrs(490) > Rrs(510)]/ Rrs(560) has wrong mathematical notation and seems copies from a NASA website. Same for the following.

Line 282: no need to be sad because two spectrometers differ from each other

Figure 4: the sharp jumps in the satellite data probably mean that the data needs some additional filtering. Also, please use superscripts when writing powers.

Figure 6: the overestimation of the OC4Me algorithm is quite logical as there is more uncorrelated CDOM than in the open ocean. I don’t think it is due to bottom effects. And still there is a decent correlation

Figure 7: CHL concentrations of less than 0.01 mg m-3 are definitely not reliable and have to be removed. The axis do not have to go beyond this range.
